# The velvet family proteins mediate low resistance to isoprothiolane in *Magnaporthe oryzae*

Fan-Zhu Meng[1], Zuo-Qian Wang[2], Mei Luo[1], Wen-Kai Wei[1], Liang-Fen Yin[3], Wei-Xiao Yin[1], Guido Schnabel[4], Chao-Xi Luo[1,3]*

1 The Hubei Key Lab of Plant Pathology, College of Plant Science and Technology, Huazhong Agricultural University, Wuhan, China, 2 Institute of Plant Protection and Soil Science, Hubei Academy of Agricultural Sciences, Wuhan, China, 3 The Experimental Teaching Center of Crop Science, College of Plant Science and Technology, Huazhong Agricultural University, Wuhan, China, 4 Department of Plant and Environmental Sciences, Clemson University, Clemson, South Carolina, United States of America

* cxluo@mail.hzau.edu.cn

**Data Availability Statement:** All relevant data are within the manuscript and its Supporting information files.

## Abstract

Isoprothiolane (IPT) resistance has emerged in *Magnaporthe oryzae*, due to the long-term usage of IPT to control rice blast in China, yet the mechanisms of the resistance remain largely unknown. Through IPT adaptation on PDA medium, we obtained a variety of IPT-resistant mutants. Based on their $EC_{50}$ values to IPT, the resistant mutants were mainly divided into three distinct categories, i.e., low resistance (LR, $6.5 \leq EC_{50} < 13.0$ μg/mL), moderate resistance 1 (MR-1, $13.0 \leq EC_{50} < 25.0$ μg/mL), and moderate resistance 2 (MR-2, $25.0 \leq EC_{50} < 35.0$ μg/mL). Molecular analysis of *MoIRR* (*Magnaporthe oryzae* isoprothiolane resistance related) gene demonstrated that it was associated only with the moderate resistance in MR-2 mutants, indicating that other mechanisms were associated with resistance in LR and MR-1 mutants. In this study, we mainly focused on the characterization of low resistance to IPT in *M. oryzae*. Mycelial growth and conidial germination were significantly reduced, indicating fitness penalties in LR mutants. Based on the differences of whole genome sequences between parental isolate and LR mutants, we identified a conserved *MoVelB* gene, encoding the velvet family transcription factor, and genetic transformation of wild type isolate verified that *MoVelB* gene was associated with the low resistance. Based on molecular analysis, we further demonstrated that the velvet family proteins VelB and VeA were indispensable for IPT toxicity and the deformation of the VelB-VeA-LaeA complex played a vital role for the low IPT-resistance in *M. oryzae*, most likely through the down-regulation of the secondary metabolism-related genes or CYP450 genes to reduce the toxicity of IPT.

## Author summary

Isoprothiolane (IPT) resistance has emerged in *Magnaporthe oryzae*, due to the long-term usage of IPT to control rice blast in China, yet the mechanisms of the resistance remain

**Funding:** This work was funded by the National Key Research and Development Program (Grant No. 2016YFD0300700) to C.L. The funders had no role in study design, data collection and analysis, decision to publish, or preparation of the manuscript.

**Competing interests:** The authors have declared that no competing interests exist.

largely unknown. Here, we explored the mechanisms of low IPT resistance in *M. oryzae*. Combining the whole genome sequencing and genetic transformation, we identified a conserved *MoVelB* gene, encoding the velvet family transcription factor to be a determinant for IPT toxicity. We further demonstrated that the deformation of the VelB-VeA-LaeA complex conferred the low IPT-resistance in *M. oryzae*, most likely through down-regulating the secondary metabolism-related genes or CYP450 genes to reduce the toxicity of IPT. This study improved our understanding of the resistance mechanism as well as the mode of action of IPT which will be helpful for making suitable strategies to manage the emerging resistance of IPT in *M. oryzae*.

## Introduction

*Magnaporthe oryzae* is the causal agent of rice blast, the most destructive disease of rice worldwide. Each year it causes significant yield losses to rice production, enough to feed 60 million people [1]. *M. oryzae*, previously known as *Magnaporthe grisea*, is a filamentous ascomycete fungus, which attacks rice at all stages of growth and infects seedlings, leaves, nodes, panicles and grains [2]. Routine use of fungicides to control crop diseases, including rice blast has become an important part of modern agriculture, helping to increase crop yields, improve quality and ensure stable production [3]. However, intensive monocropping and long-term usage of single-site fungicides provide an favorable environment for the development of fungicide resistance [4].

Isoprothiolane (IPT), an agricultural fungicide developed by Nihon Nohyaku Co., Ltd. in 1974, is a systemic fungicide in the dithiolanes class with both protective and curative effects for controlling rice blast disease [5]. In China, IPT has been routinely used to control leaf blast and panicle blast for 1–3 times a season. It is not only effective in controlling rice blast, but is also used as insecticide to control rice plant hopper [6], and as plant-growth regulator to promote the growth of plant roots [7]. IPT inhibits the methylation of phosphatidylethanolamine to phosphatidylcholine, and is thus considered as a choline biosynthesis inhibitor with cross resistance with organophosphorus fungicides such as iprobenfos [8, 9]. Resistance to IPT (also known as Fuji One) in *M. oryzae* has already been detected in rice fields in recent years [10], but little is known about the molecular basis of resistance. Lab IPT-resistant mutants did not possess changes in the DNA sequence and expression level of genes related to the synthesis of phosphatidylcholine [11], indicating the existence of resistance mechanisms unrelated to the putative target enzyme. Interestingly, mutations in a $Zn_2Cys_6$ transcription factor MoIRR, resulted in a moderate level of resistance to IPT in *M. oryzae* [12]. Our research group found that most of the IPT resistance in field isolates were unstable, and only showed low resistance (LR). Thus, we adapted the wild type isolate H08-1a on PDA amended with IPT in a series of concentrations and obtained a variety of resistant mutants with different levels of resistance. The objective of this study was to explore the resistance mechanisms of low resistance to IPT in *M. oryzae*.

## Results

### Induction of IPT-resistant mutants in *M. oryzae*

To explore the mechanism of IPT resistance, the wild type isolate H08-1a was adapted on PDA amended with IPT. We found that the proportion of stable resistant mutants increased with increasing concentrations of IPT (Fig 1A). A total of 77 IPT-resistant mutants were obtained

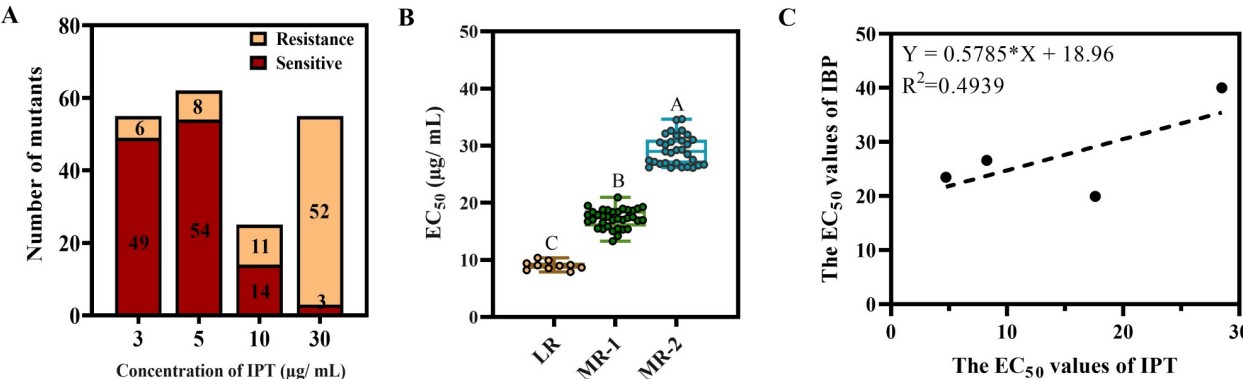

**Fig 1. Isoprothiolane taming of *M. oryzae*.** (**A**) Proportion of stable IPT resistant mutants at different taming fungicide concentrations. (**B**) The IPT resistant mutants exhibited significant differences in their resistance to IPT. Data presented are the mean ± SD (n = 3). Bars followed by the same letter are not significantly different according to a LSD test at P = 0.01. (**C**) Detection of cross-resistance to IBP in IPT resistant mutants. Linear regression analysis was performed with $EC_{50}$ values of IPT and IBP for IPT-resistant mutants.

with $EC_{50}$ values between 7.5–45 μg/mL. Except for the mutant 30–50 which was considered as highly resistant (HR; $EC_{50}$ value 43.55 μg/mL), the resistant mutants were divided into three distinct categories, i.e., LR ($6.5 \leq EC_{50} < 13$ μg/mL), MR-1 ($13 \leq EC_{50} < 25$ μg/mL), and MR-2 ($25 \leq EC_{50} < 35$ μg/mL) containing 10, 35, and 31 mutants, respectively. There were significant differences of $EC_{50}$ values between different resistance categories (Fig 1B). Even at low dose IPT exposure, multiple phenotypic resistant mutants readily emerged, indicating a high risk of resistance emergence upon frequent exposure to IPT in fields.

Positive cross-resistance between IPT and iprobenfos (IBP) has been established in the past. Thus, we determined the sensitivity of the resistant mutants to IBP to clarify whether the IPT mutants showed resistance to IBP. Unexpectedly, the R-value of the linear regression of the $EC_{50}$ values of IPT and IBP was 0.4939, indicating that the IPT-resistant mutants were not significantly cross-resistant to IBP (Fig 1C). These results suggested that IPT and IBP may not have the same mode of action.

### Fitness assessment of IPT-resistant mutants

To assess the fitness of IPT-resistant mutants, mycelial growth rate, melanin production, sporulation, conidial germination and virulence were evaluated. Compared with the wild-type isolate H08-1a, the LR mutant showed significantly reduced mycelial growth, conidial germination, increased sporulation, and increased melanin production (Fig 2A–2D), while other mutants did not show significant changes for any of the investigated phenotypes (Fig 2). The virulence of the LR mutant was comparable with parental isolate and the other mutants (Fig 2). These results suggest that mutants with different IPT phenotypes possess different resistance mechanisms.

To further characterize IPT-resistant mutants, we determined the sensitivity of different mutants to seven common fungicides with different modes of action. IPT-resistant mutants were more sensitive than isolate H08-1a to fungicides associated with osmotic pathway regulation, i.e., the antibiotics rapamycin (RAP), PP fungicide fludioxonil (FLU), and DCF fungicide iprodione (IPR), although each mutant had specific sensitivity profiles (Fig 2F). The resistant mutants did not differ in their sensitivity to other fungicides when compared to parental isolate H08-1a (Fig 2F).

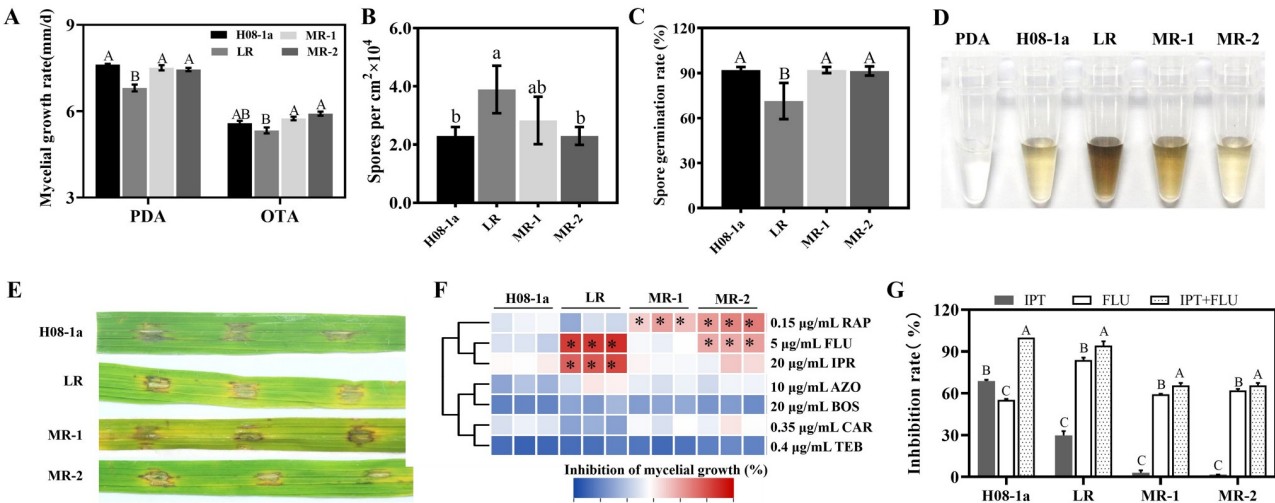

**Fig 2. Assessment of fitness of different IPT resistant mutants.** (**A**) Growth rate of IPT resistant mutants on PDA and OTA. (**B**) Sporulation of different IPT resistant mutants. (**C**) Conidial germination of different IPT resistant mutants. (**D**) Detection of melanin accumulation in IPT resistant mutants. (**E**) Pathogenicity test of different IPT resistant mutants. (**F**) Mycelial inhibition rate heat map of IPT resistant mutants to different fungicides. H08-1a and IPT resistant mutants were inoculated on PDA or PDA amended with 0.15 μg/mL rapamycin (RAP), 5 μg/mL fludioxonil (FLU), 20 μg/mL iprodione (IPR), 10 μg/mL azoxystrobin (AZO), 20 μg/mL boscalid (BOS), 0.35 μg/mL carbendazim (CAR), 0.4 μg/mL tebuconazole (TEB). (**G**) Control efficacy of IPT resistant mutants. H08-1a and IPT resistant mutants were inoculated on PDA or PDA amended with 5 μg/mL IPT, 5 μg/mL FLU, combination of 5 μg/mL IPT and 5 μg/mL FLU. Data presented are the mean ± SD (n = 3). Bars followed by the same letter are not significantly different according to a LSD test at P = 0.01.

Fludioxonil has been shown to control IPT-resistant *MoIRR* mutants [13]. In this study, we further characterized the response of IPT-resistant mutants cultured on 5 μg/ mL IPT and 5 μg/ mL FLU. Results showed that FLU could effectively inhibit the mycelial growth of all types of resistant mutants, and the combination of IPT and FLU had better control efficacy than IPT or FLU alone (Fig 2G).

## Mutations in *MoIRR* were mainly associated with moderate IPT-resistance

The *MoIRR* (*MGG_04843*) gene was sequenced from all 77 IPT resistant-mutants. It was found that 31 mutants containing *MoIRR* mutations showed moderate resistance to IPT and grouped into the MR-2 resistance phenotype (Fig 3A and 3B), while the LR mutants did not reveal mutations in *MoIRR* gene (S1 Table). MR-2 mutants accounted for 40% of all resistant mutants, and the main types of mutation in *MoIRR* gene were frameshifts and substitutions (Fig 3C). The mutation sites were located in exons and introns, but they were mainly concentrated in the fourth exon that corresponded to the Fungal_TF_MHR domain (fungal transcription factor regulatory middle homology region) (Fig 3D and 3E). Nevertheless, resistance to IPT was similar among *MoIRR* mutants with different mutation types and that mutations in *MoIRR* caused moderate but not low IPT resistance in *M. oryzae*.

## The velvet family protein MoVelB was associated with the IPT low resistance

To further explore the mechanism of low IPT resistance in *M. oryzae*, we performed whole-genome sequencing, then SNP and InDel analysis for the LR mutant 3–15. As shown in Table 1, we obtained 6 candidate IPT-resistance genes, including three transcription factor encoding genes *MGG_01620*, *MGG_01870*, and *MGG_02962*, a metalloproteinase encoding

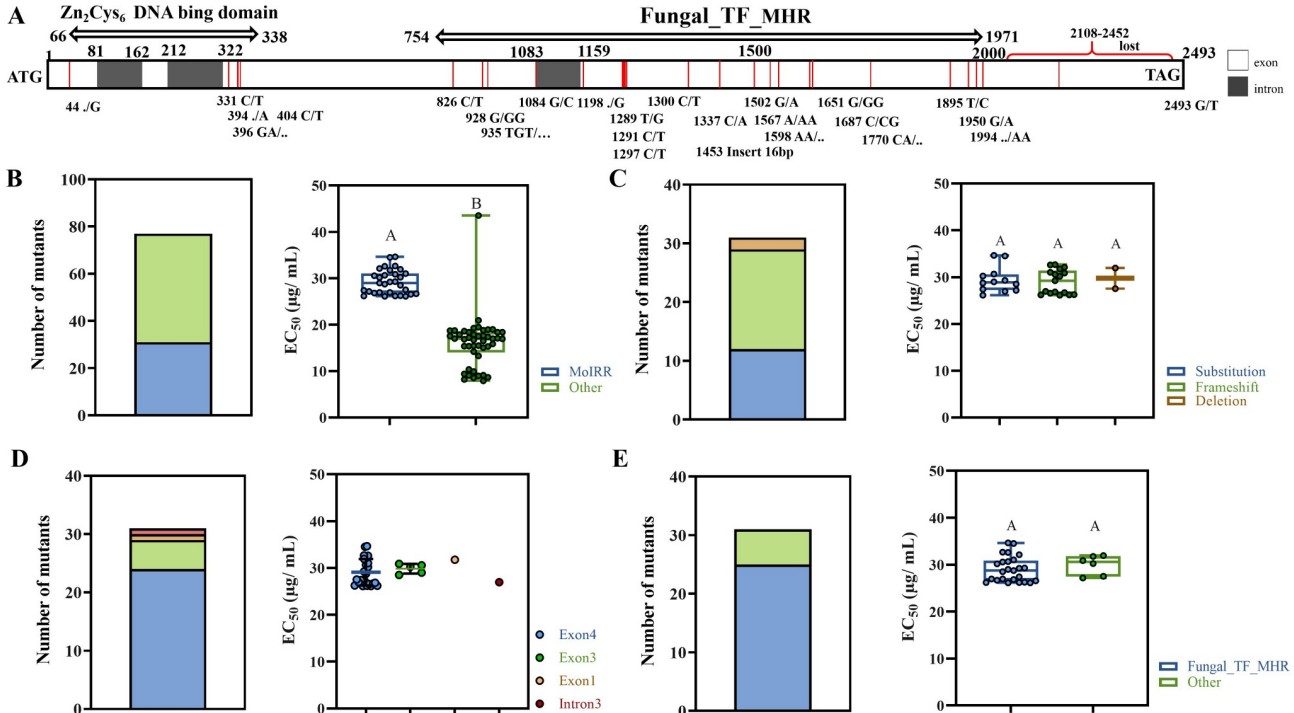

**Fig 3. Mutation of *MoIRR* was one of the main causes of IPT resistance in *M. oryzae*.** (**A**) Mutations of *MoIRR* gene in IPT resistant mutants. (**B**) Percentage of mutants with *MoIRR* mutations in all resistant mutants. (**C**) Mutation types in MR-2 mutants. (**D**) Mutation sites of *MoIRR* gene in MR-2 mutants. (**E**) Distribution of mutations in different domains in MR-2 mutants. Bars followed by the same letter are not significantly different according to a LSD test at P = 0.01.

gene *MGG_15370*, a linker histone family protein encoding gene *MGG_12797*, and a helicases encoding gene *MGG_16993*.

*MGG_01620* encodes the developmental regulation protein VelB, which usually forms a complex with members of the velvet factor family and is involved in fungal growth, development and secondary metabolism. It was found that the CAG (glutamine) at the 305th codon (Q305X) of the *MoVelB* gene was substituted to the terminator (TAG) in the LR mutant 3–15 (Fig 4A). Phylogenetic analysis showed that VelB was conserved in filamentous fungi and contained a DNA binding velvet domain (Fig 4B). Further investigation of other LR mutants showed that *MoVelB* mutations could be detected from 60% of the LR mutants and that the main types of mutation were frameshifts (S1F Fig).

To explore the resistance mechanisms in the LR mutants, we generated knockout transformants of six candidate genes and investigated their corresponding resistance phenotypes.

**Table 1. Identification of candidate genes for low IPT resistance in *M. oryzae***

| Gene ID | Type of mutation | cDNA changes | Protein changes | Gene description |
|---|---|---|---|---|
| MGG_01620 | stop gain | C913T | Q305X | Developmental regulator VelB |
| MGG_01870 | nonsynonymous SNP | G706A | V236I | $Zn_2Cys_6$ fungal-type DNA-binding domain |
| MGG_16993 | nonsynonymous SNP | G2523A | M841I | DEAD/DEAH box helicase domain |
| MGG_12797 | nonsynonymous SNP | G69T | Q23H | Linker histone H1/H5 |
| MGG_15370 | nonsynonymous SNP | G583C | D195H | Metalloproteinase |
| MGG_02962 | frameshift deletion | 784delC | 263Rfs*186 | $Zn_2Cys_6$ fungal-type DNA-binding domain |

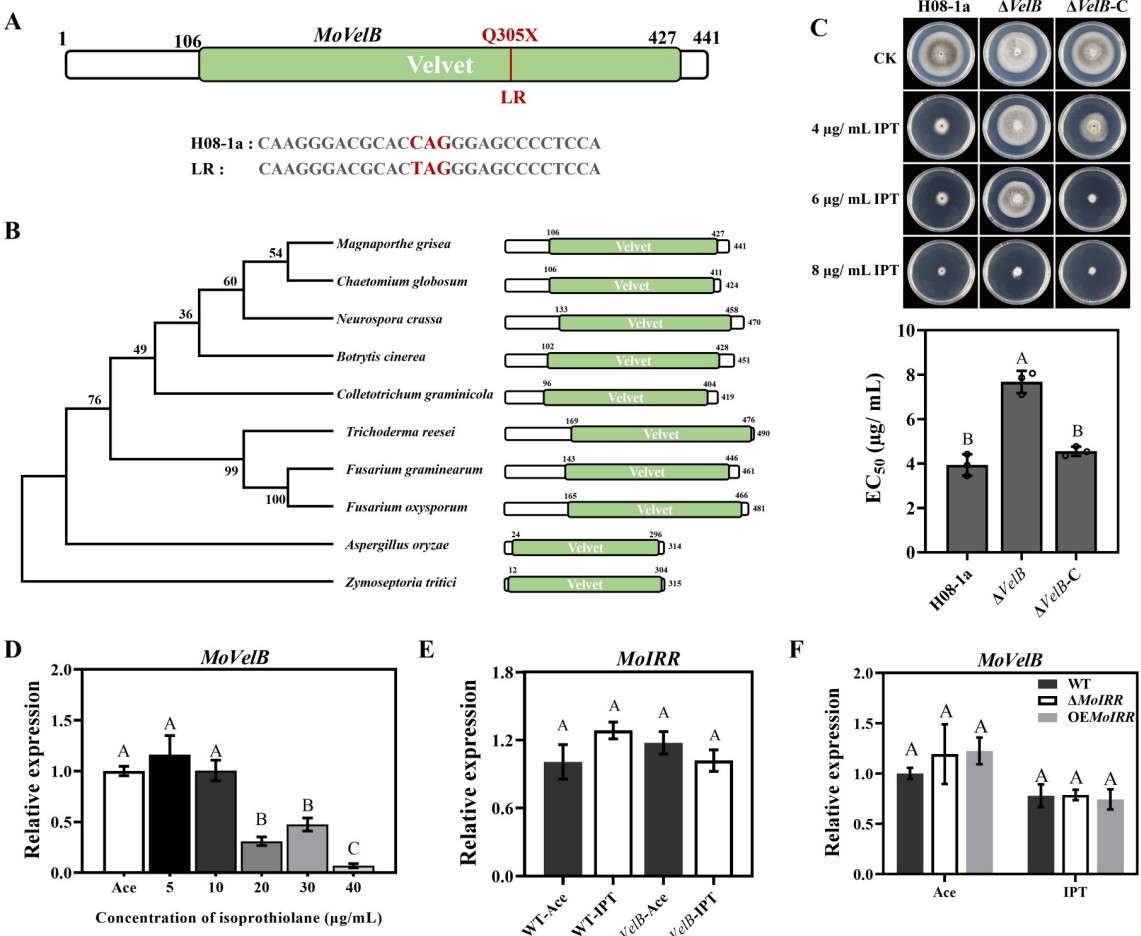

**Fig 4. MoVelB negatively regulated the low resistance to IPT in *M. oryzae*.** (**A**) Identification of IPT resistance loci in the LR mutant 3–15. (**B**) Phylogenetic tree of MoVelB homologous proteins. The amino acid sequences of proteins in phylogenetic tree were retrieved from the NCBI database. Domains were aligned with ClustalW, and the tree was constructed with the neighbor-joining method. The VelB velvet domain was indicated as green boxes. (**C**) Knockout transformant ΔVelB (ΔVelB-2) displayed low resistance to IPT. A 3-mm mycelial plug of each strain was inoculated on PDA or PDA amended with 4, 6, 8 μg/ mL IPT and then incubated at 27˚C for 5 days (top panel), and the mycelial growth inhibition was calculated for each strain (below panel). (**D**) Expression of *MoVelB* in wild type isolate H08-1a at different concentrations of IPT. (**E**) Expression of *MoIRR* in different strains with or without IPT. (**F**) Expression of *MoVelB* in wild type isolate H08-1a, *MoIRR* knockout (ΔIRR-1) and overexpression (OEIRR-1) transformants with or without IPT. Ace: acetone, the solvent of IPT; IPT: isoprothiolane. The *MoActin* gene was used as the internal reference for normalization. Data presented are the mean ± SD (n = 3). Bars followed by the same letter are not significantly different according to a LSD test at P = 0.01.

*MoVelB* (*MGG_01620*) knockout transformants exhibited significantly increased IPT resistance, just like the resistance level in LR mutant 3–15 (Figs 4C and S1B), transformants of other candidate genes did not show significantly different sensitivity to IPT compared to parental isolate H08-1a.

Transformation of the 2k promoter region and the coding region of *MoVelB* into the knockout transformant ΔVelB-2 restored the IPT sensitivity to wild type level (Fig 4C), in addition, OE*VelB* transformants increased sensitivity to IPT (S1D Fig), indicating that MoVelB negatively regulates the low IPT resistance in *M. oryzae*.

To further clarify the molecular mechanism by which MoVelB negatively regulates the resistance to IPT in *M. oryzae*, we examined the expression of *MoVelB* when subjected

different concentrations of IPT together with impact on *MoIRR* expression, a gene related to moderate IPT resistance. Results showed that the expression of *MoVelB* was significantly suppressed at high concentrations of IPT (Fig 4D), and that the knockout of *MoVelB* did not affect the expression of *MoIRR* (Fig 4E). Neither the knockout nor the overexpression of *MoIRR* affected the expression of *MoVelB* (Fig 4F), indicating that the down regulation of *MoVelB* upon IPT exposure is a novel mechanism, different from the MoIRR regulatory pathway.

## Biological functions of MoVelB in *M. oryzae*

To investigate whether MoVelB is involved in the growth and development of *M. oryzae*, the colony morphology of the different strains were investigated, and the mycelial growth rates were measured. Compared to parental isolate H08-1a, knockout transformants showed fluffy colonies (S2A Fig), produced fewer conidia, and the mycelial growth rate decreased on the CM, MM, PDA or OTA media. The complemented transformants restored the colony growth rate and sporulation (Fig 5A and 5B). These results indicated that MoVelB played important roles in regulating mycelial growth and sporulation. MoVelB knockout transformants produced significantly more melanin when cultured in PDB for 6 days compared to the parental isolate or the complemented transformant (Figs 5C and S2B), indicating MoVelB plays a negative regulation role in the synthesis of melanin.

To determine whether MoVelB is involved in response to environmental stresses, H08-1a, Δ*VelB*, Δ*VelB*-C were inoculated onto PDA plates amended with different stress agents, including osmotic stresses NaCl, KCl and SOR, cell wall/cell membrane stresses SDS, CR and CFW, and oxidative stress $H_2O_2$. After incubation at 27˚C for 5 days, Δ*VelB* showed a decreased tolerance to $H_2O_2$ and increased tolerance to SOR, but no difference was observed upon other stresses in comparison to the wild-type H08-1a and Δ*VelB*-C transformant (Figs 5D and S2C). Catalase gene *Cat3* plays an important role in oxidative stress response. As expected, the expression of *Cat3* was significantly reduced in Δ*VelB* transformants (Fig 5E), indicating that MoVelB plays an important role in response to oxidative stress by regulating the expression of *Cat3* in *M. oryzae*.

To further characterize whether MoVelB-related IPT-resistant mutants are also resistant to other fungicides, we tested the sensitivity of Δ*VelB* transformants to iprobenfos (IBP), fludioxonil (FLU), iprodione (IPR), tebuconazole (TEB), azoxystrobin (AZO), boscalid (BOS), carbendazim (CAR). Compared to parental isolate H08-1a and complemented transformant Δ*VelB*-C, Δ*VelB* showed resistance to the organophosphorus fungicide IBP (Figs 5F and S2D). In addition, Δ*VelB* was remarkably sensitive to PP fungicide FLU and DCF fungicide IPR (Figs 5F and S2D). FLU and IPR are thought to stimulate MAP kinases in osmotic signaling, leading to excessive glycerol accumulation and consequent cell death. As shown in the Fig 5G and 5H, we found that MoVelB was involved in FLU and IPR tolerance by regulating the expression of *Gpd1* and *Gpp1* which are participated in glycerol synthesis in *M. oryzae*.

## Velvet family proteins were indispensable for toxicity of IPT

The velvet family proteins include MoVeA, MoVelB, MoVelC and MoVosA in *M. oryzae*, all of which have a conserved velvet domain (Fig 6A). The velvet complex is a class of global transcription factors involved in secondary metabolism, growth and development in filamentous fungi. To characterize the functional roles of velvet genes in IPT-resistance of *M. oryzae*, corresponding knockout transformants Δ*VeA*, Δ*VelC*, and Δ*VosA* were generated through PEG mediated protoplast transformation. Results showed that the IPT-resistance in Δ*VeA* was comparable to that of Δ*VelB* and that Δ*VosA* and Δ*VelC* also showed resistance to IPT, but at lower levels than Δ*VelB* (Fig 6B). Meanwhile overexpression transformants OE*VeA* increased the

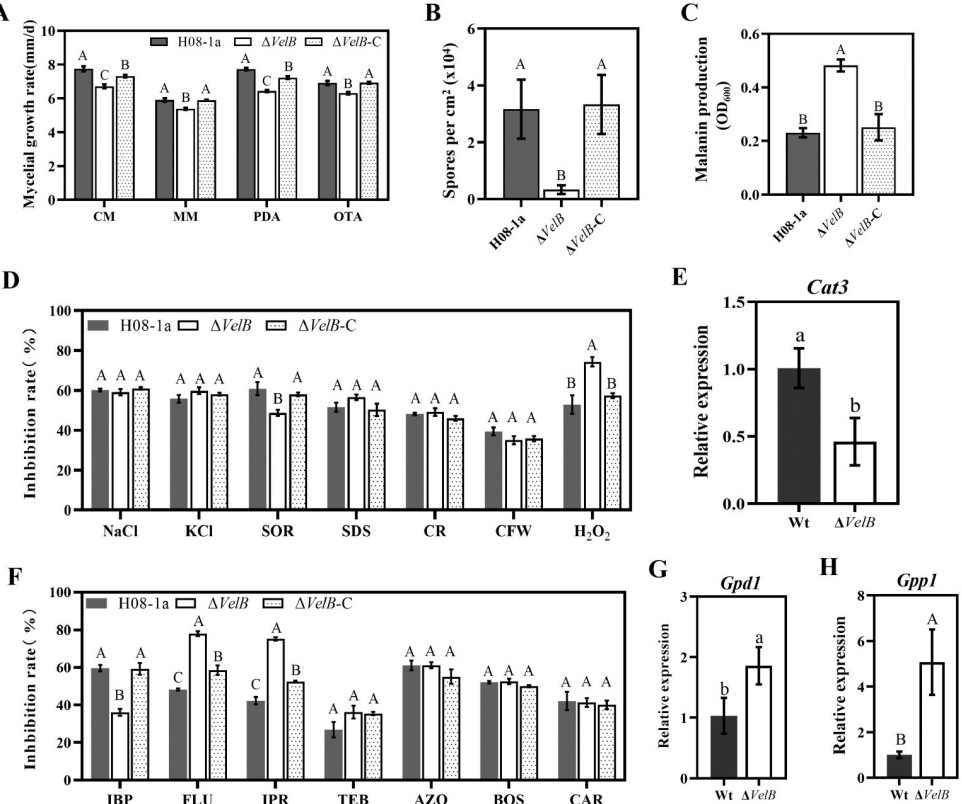

**Fig 5. MoVelB regulated mycelial growth, sporulation, melanin synthesis, and oxidative stress.** (**A**) Comparisons in colony morphology among the parental isolate H08-1a, Δ*VelB* and the complemented transformant Δ*VelB*-C grown on CM, MM, PDA, or OTA, mycelial growth rate was calculated accordingly. (**B**) Sporulation of different types of strains. (**C**) Production of melanin of different strains cultured in PDB for 7 days. (**D**) The H08-1a, Δ*VelB* and Δ*VelB*-C strains were incubated on PDA amended with different stress agents at 27˚C for 5 days and statistical analysis of the growth inhibition rate. (**E**) RT-qPCR analyses of the expression of *Cat3* in Δ*VelB*, compared to H08-1a. The *MoActin* gene was used as the internal reference for normalization. (**F**) The H08-1a, Δ*VelB* and Δ*VelB*-C strains were incubated on PDA amended with different fungicides at 27˚C for 5 days and statistical analysis of the growth inhibition rate. (**G, H**) RT-qPCR analyses of the expression of *Gpd1*, *Gpp1* in Δ*VelB*, compared to H08-1a. The *MoActin* gene was used as the internal reference for normalization. Data presented are the mean ± SD (n = 3). Bars followed by the same letter are not significantly different according to a LSD test, lowercase letters indicate the p-value <0.05 and uppercase letters indicate the p-value <0.01.

sensitivity to IPT (Fig 6C and 6D). It was rather intriguing to note that Δ*VeA* also showed reduced tolerance to H₂O₂, FLU and IPR and increased resistance to IBP, similar with that appeared in Δ*VelB* (S3 Fig). These results suggest that MoVeA and MoVelB may jointly be involved in the regulation of IPT toxicity and oxidative stress in *M. oryzae*.

The expression pattern of *MoVeA* under IPT stress was determined by using RT-qPCR. In contrast to *MoVelB*, the *MoVeA* expression was induced by IPT (Fig 6F). The interaction between MoVelB and MoVeA for IPT toxicity was further demonstrated by using the Co-IP and yeast two-hybrid method. The results showed that MoVelB interacted with MoVeA (Figs 6G and S4A). At the same time, *MoVelB* and *MoVeA* double knockout transformants Δ*VelB*Δ-*VeA* were obtained (S4B Fig) which also showed low resistance to IPT, and the level of resistance was not significantly different from Δ*VelB* or Δ*VeA* (Fig 6H), suggesting that MoVelB determined IPT toxicity by a direct interaction with MoVeA.

The nuclear protein LaeA can form heterotrimer VelB/VeA/LaeA with the VelB and VeA complex in the dark to co-regulate secondary metabolism, growth and development processes

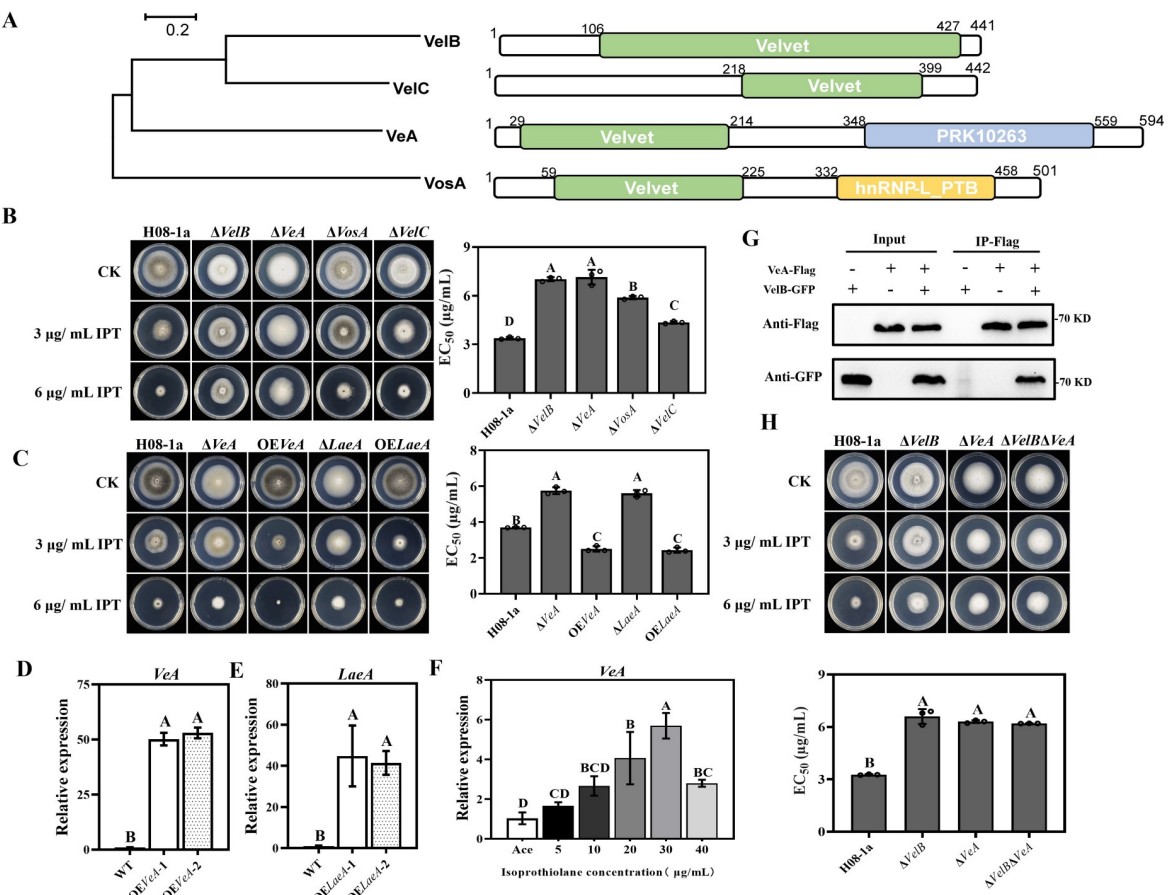

**Fig 6. Involvement of velvet family proteins in resistance to IPT in *M. oryzae*.** (**A**) Velvet family proteins contain a velvet structural domain identified using the NBCI protein database (https://www.ncbi.nlm.nih.gov/cdd). (**B**) Knockout of velvet family proteins encoding genes led to reduced sensitivity to IPT. (**C**) Sensitivity of MoVeA and MoLaeA-related mutants to IPT. (**D**) Detection of *MoVeA* expression in OE*VeA* transformants by RT-qPCR. (**E**) Detection of *MoLaeA* expression in OE*LaeA* transformants by RT-qPCR. (**F**) Expression of *MoVeA* at different concentrations of IPT. The *MoActin* gene was used as the internal reference for normalization. (**G**) The Co-IP assay revealed that MoVelB interacted directly with MoVeA. (**H**) *MoVelB* and *MoVeA* double knockout transformants reduced low resistance to IPT. Data presented are the mean ± SD (n = 3). Bars followed by the same letter are not significantly different according to a LSD test at P = 0.01.

in *Aspergillus nidulans* and *MoLaeA* (*MGG_01233*) has been reported to be involved in the regulation of tenuazonic acid synthesis in *M. oryzae*. In the current study, yeast two-hybrid assay showed that MoLaeA could directly interact with MoVeA, demonstrated that MoLaeA might form a heterotrimer complex with the MoVeA and MoVelB through direct interactions (S4C Fig). Knockout of *MoLaeA* caused defects in vegetative growth and colony pigmentation on PDA compared with the parental isolate, and Δ*LaeA* exhibited low resistance to IPT, comparable to resistance levels found for Δ*VeA* and Δ*VelB*, in addition, OE*LaeA* transformants increased sensitivity to IPT (Fig 6C and 6E). To clarify that the VelB/VeA/LaeA complex was the key to regulate low IPT resistance, double knockout transformants Δ*VeA*Δ*LaeA*, and triple knockout transformants Δ*VelB*Δ*VeA*Δ*LaeA* were obtained (S4D Fig). As shown in S4E Fig, Δ*VeA*Δ*LaeA* and Δ*VelB*Δ*VeA*Δ*LaeA* also showed low resistance to IPT. These results suggested that the disruption of the VelB/VeA/LaeA complex was involved in the low IPT-resistance in *M. oryzae*.

## Identification of a regulatory network of VelB-VeA-LaeA complex

To further verify the involvement of VelB-VeA-LaeA complex for the IPT toxicity in *M. oryzae* through the regulation of secondary metabolism, genome-wide expression of wild-type H08-1a and Δ*VelB* responding to IPT was analyzed by RNA-seq. There were 1717 differentially expressed genes (DEG) in Δ*VelB*, of which 657 were up-regulated and 1060 were down-regulated, 4 h after IPT treatment (Fig 7A). Analysis of the Kyoto Encyclopedia of Genes and Genomes (KEGG) pathway showed that the DEGs were mainly enriched in the metabolic pathway, with the most significant enrichment in the synthesis of secondary metabolites (Fig 7B). Heat map with the DEGs (Log$_2$FC(Δ*VelB*_RPKM/H08-1a_RPKM)) > 2 or < -2) in biosynthesis of secondary metabolites was shown in Fig 7C and S4 Table. Meanwhile, RT-qPCR confirmed that some DEGs in Δ*VelB* showed similar expression patterns in Δ*VeA* and Δ*LaeA*, indicating that the VelB-VeA-LaeA complex significantly regulated secondary metabolism-related genes under IPT pressure. Particularly, five DEGs, i.e., *Mo4HNR*, *MoGph1*, *MoGad1*, *MoAkr1*, and *MoLpp1* were down regulated in Δ*VelB*, similar results were observed in Δ*VeA* and Δ*LaeA* (Fig 7D). As expected, these genes were up regulated in OE*VelB* (S5A Fig). These data suggested that the disruption of VelB-VeA-LaeA complex determined the low IPT

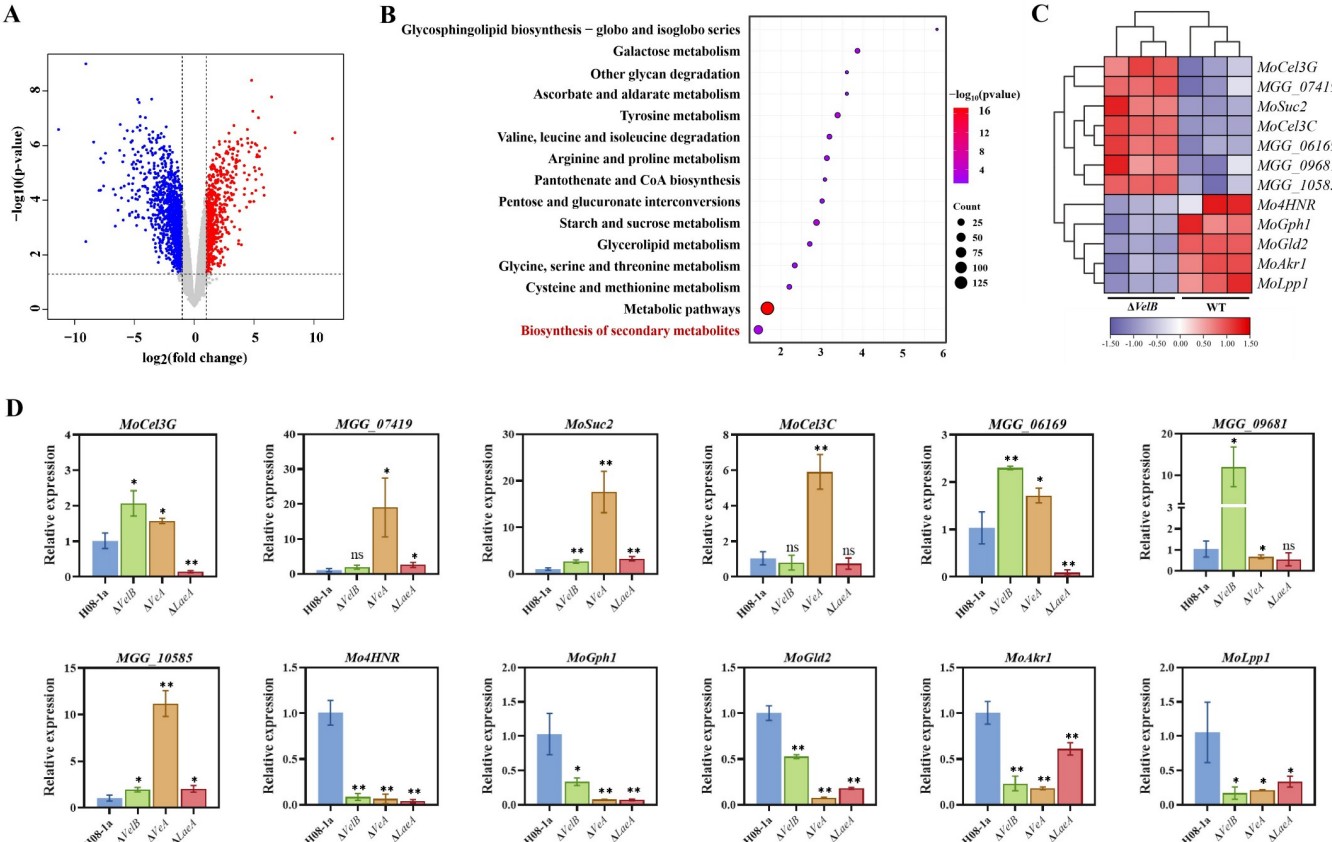

**Fig 7. Involvement of MoVelB-MoVeA-MoLaeA complex in IPT resistance was possibly through regulation of secondary metabolism in *M. oryzae*.** (**A**) Volcano plot analysis of DEGs (log$_2$(fold change ≥1 or ≤ -1 and P value of ≤ 0.05) in Δ*VelB*, compared to H08-1a. (**B**) KEGG enrichment analysis of DEGs in Δ*VelB* under IPT pressure. (**C**) Head map analysis of DEGs (log$_2$(fold change ≥1.5 or ≤ -1.5)) in biosynthesis of secondary metabolites. (**D**) Regulation of DEGs in biosynthesis of secondary metabolites by MoVelB, MoVeA, MoLaeA determined by RT-qPCR. Data presented are the mean ± SD (n = 3). Statistical significance was determined using Student's t test with a two-tailed distribution and two-sample unequal variance, *, P ≤ 0.05; **, P ≤ 0.01.

resistance, most likely through down-regulating secondary metabolism-related genes to reduce the toxicity of IPT.

## Discussion

IPT, tricyclazole and azoxystrobin are the most important chemicals for controlling rice blast in China. As an effective and low-toxic fungicide, IPT is used for the prevention and treatment of rice blast. However, it can easily bring about the emergence of resistance under long-term usage, and the frequency of resistance to IPT resistant isolates has been more than 50% in Liaoning Provinces in China [14]. In this study, we investigated the mechanism of resistance and assessed the risk of resistance to IPT in *M. oryzae*.

Very little is known about the molecular basis of IPT resistance in *M. oryzae* or any other plant pathogenic fungus. The $Zn_2Cys_6$ transcription factor MoIRR was shown to be associated with IPT resistance in *M. oryzae* due to the frameshifts and SNPs in the *MoIRR* gene [12]. In this study, the IPT resistance mechanisms were further investigated. It was found that the *MoIRR* point mutants including Q48Stop, A72V, H213Y, L346F, R413H, F544S, W562Stop, Stop743Y confer moderate resistance. In mutants with low resistance, no *MoIRR* mutations were observed, indicating that other mechanisms determine low IPT resistance in *M. oryzae*.

In recent years, isolates with low IPT resistance and lacking variations in the *MoIRR* gene, have been reported in Liaoning and Jiangsu provinces in China [10, 14]. Meanwhile we identified a large number of low resistance mutants without mutations in *MoIRR*. Therefore, it is important to explore the molecular mechanisms of low resistance in *M. oryzae*. Based on whole-genome sequencing, the velvet transcription factor MoVelB was identified to be associated with the low resistance in the LR mutants. Genetic transformation proved that knockout of *MoVelB* gene resulted in the development of resistance. It has been reported that VelB plays crucial roles in the fungal development, production of conidia and pigments in *A. nidulans* [15], *B. cinerea* [16], *C. lunata* [17], *M. oryzae* [18], *V. dahlia* [19], *V. mali* [20]. In addition, VelB is indispensable for virulence in several filamentous phytopathogenic fungi [21]. For the first time, we found that MoVelB negatively regulated low IPT resistance, and the regulatory mechanism of MoVelB on IPT resistance is different from the regulatory pathway of MoIRR which confers moderate resistance in *M. oryzae* [12].

Velvet family proteins usually function as homodimers or heterodimers. Nuclear entry of VelB requires the formation of a heterodimer with VeA and the assistance of α-importin KapA protein. In the dark, VeA-VelB heterodimers enter the nucleus, promote sexual fruiting body formation [22]. VelB also forms homodimers and participates in asexual reproduction [23]. LaeA can bind VeA and VelB to form heterodimers that participate in secondary metabolism [24, 25]. We also found that ΔVeA and ΔLaeA showed low resistance to IPT, with similar levels of resistance to ΔVelB. In addition, double and triple knockout transformants ΔVeAΔLaeA and ΔVelBΔVeAΔLaeA also showed low resistance to IPT. Furthermore, we verified that MoVeA could interact with MoVelB and MoLaeA, respectively, but MoVelB could not interact with MoLaeA. Thus, it was proposed that the heterotrimer VelB-VeA-LaeA was generated through MoVeA interacted with MoVelB and MoLaeA, respectively. Through the disruption of the heterotrimer formation, the downstream genes involving secondary metabolisms were down regulated to reduce the toxicity of IPT and caused the corresponding low IPT resistance in *M. oryzae* (Fig 8).

NF-κB is a protein heterodimer consisting of p50 and RelA, which could be activated by many types of cellular stresses, leading to the promotion of cell survival [26]. The velvet proteins with a DNA binding domain structurally similar to that of NF-κB p50, VelB and VeA heterodimer can activate p450 BapA expression in response to stimulation of the five-ring PAH

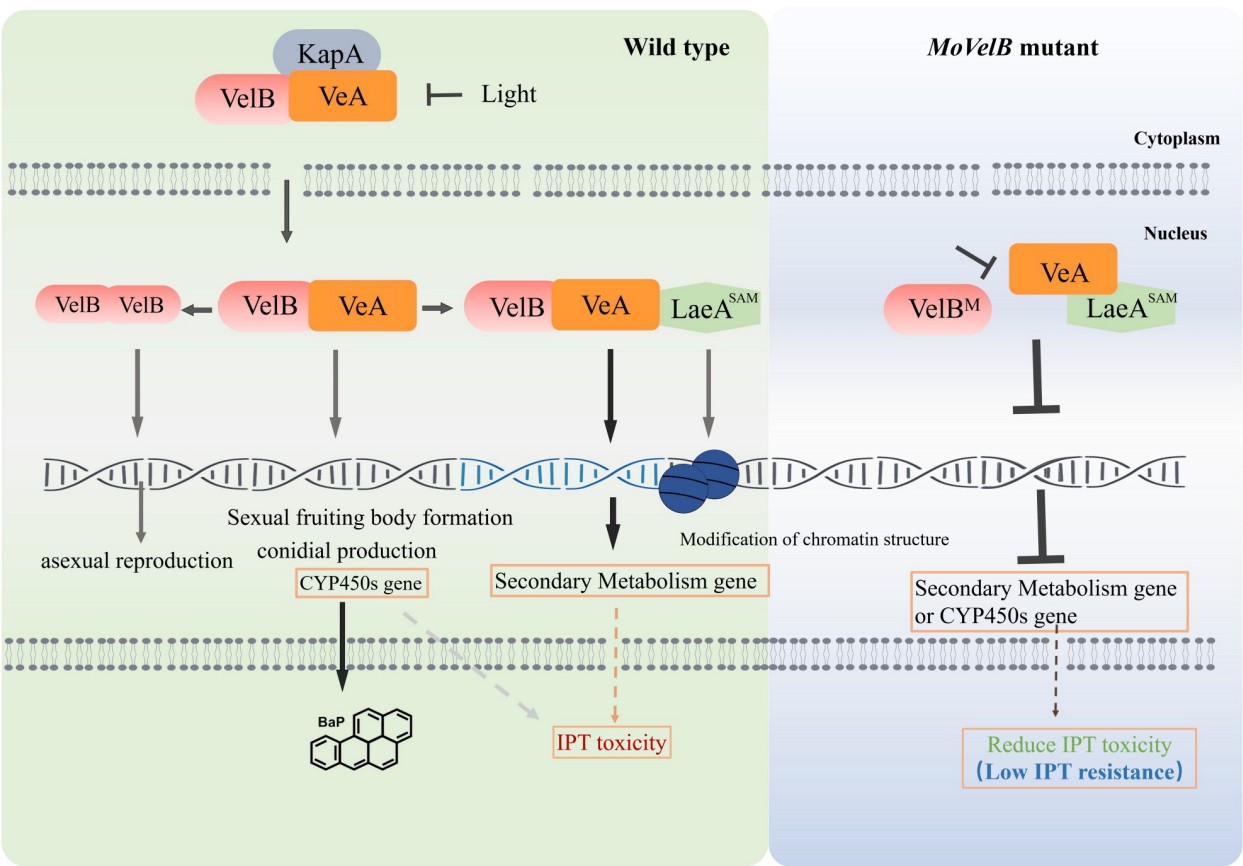

**Fig 8. Model of velvet family proteins in regulating low IPT resistance in *M. oryzae*.** VelB, VeA, LaeA, and KapA indicate the MoVelB, MoVeA, MoLaeA, and MoKapA, respectively.

BaP (benzo[a]pyrene) in *A. nidulans* [27]. Interestingly, DEGs with monooxygenase activity were highly enriched by Gene Ontology analysis, with 15 p450 proteins being expressed by MoVelB activation under IPT stress (S5B and S5C Fig and S5 Table), suggesting that the regulation of low IPT resistance may also be realized through similar mechanism (Fig 8).

In conclusion, we explored the mechanisms involved in the low IPT resistance in *M. oryzae*. Genomic sequencing combined with genetic transformation demonstrated that the low resistance was negatively regulated by the *MoVelB* gene. Further molecular analysis showed that another velvet protein MoVeA interacted with MoVelB and MoLaeA to form the VelB-LaeA-VeA complex, which regulated the genes involving secondary metabolisms to achieve the toxicity of IPT, disruption of the VelB-LaeA-VeA complex conferred the low IPT resistance, most likely through down-regulating the secondary metabolism related genes to reduce the toxicity of IPT in *M. oryzae*.

## Materials and methods

### Mutants, media and fungicides

The *M. oryzae* strains including wild type isolate H08-1a and resistant mutants are listed in S1 Table. All mutants were cultured on potato dextrose agar (PDA) medium for 5 days at 27˚C in the dark. For vegetative growth, 3 mm × 3 mm mycelial plugs from the periphery of freshly

cultured mutants were inoculated onto complete medium (CM), minimal medium (MM), PDA or Tomato Oat Agar (OTA). The fungicide IPT was dissolved in acetone to make stock solution at the concentration of 8000 μg a.i. /mL. Sensitivity to IPT was assessed on PDA amended with IPT at 0, 1, 2, 5, 10, 30 and 50 μg/mL. Growth inhibition was calculated, and regression against the logarithm of fungicide concentrations was analyzed to obtain $EC_{50}$ values. *Escherichia coli* JM109 was grown in LB medium with ampicillin (100 μg/mL) for plasmid amplification. Sensitivity to rapamycin (Rap), fludioxonil (FLU), iprodione (IPR), tebuconazole (TEB), azoxystrobin (AZO), boscalid (BOS), carbendazim (CAR) was assessed on PDA amended with corresponding fungicides at concentrations 0.15, 5, 20, 0.4, 10, 20, 0.35 μg/mL, respectively.

### Induction of IPT resistant mutants

The hyphal plugs (3 mm in diameter) of wild-type isolate H08-1a were inoculated on PDA plates containing different concentrations of IPT at 3, 5, 10, or 30 μg/mL, where 300 hyphal plugs were inoculated at each concentration. Mycelium fanning out from inoculation plugs situated on IPT-amended medium was transferred for 6 consecutive generations (one generation: six days on fungicide free PDA followed by another six days on IPT-amended PDA). Strains that can stably grow on IPT-amended PDA were considered as IPT resistant mutants.

### Sequence analysis of *MoIRR* gene in resistant mutants

*MoIRR* gene was identified previously to be associated with IPT resistance [12]. Therefore, the primer pair MoIRR-Check-F/R was designed based on the MG8 genome to amplify the full-length of *MoIRR* gene in resistant mutants to detect the presence of mutations in *MoIRR* (S2 Table).

### Whole genome sequencing

Cetyltrimethylammonium bromide (CTAB) method was used to extract genomic DNA from the LR mutant 3–15. Genome sequencing was conducted on the Illumina HiSeq 4000 PE150 platform using 150 bp paired-end libraries with 500 bp inserts at Wuhan SeqHealth Technology Company. We used Lofreq (version 2.1.5) software to perform SNP and InDel assays. In order to reduce unnecessary mutation sites, we specified strict screening conditions: 1. The mutation appears in the open reading frame region; 2. The mutation can only appear once in a gene; 3. The candidate gene is expressed in the hyphal growth stage; 4. The candidate gene is not mutated in the 1a_mut genome because the colony morphology of 3–15 is completely different from that of 1a_mutant, a mutant with moderate resistance to IPT.

### Phylogenetic analysis

The amino acid sequences of the VelB and BLAST Servers at NCBI from *Magnaporthe* genome (https://www.ncbi.nlm.nih.gov/genome/51706) were used. Phylogenetic trees were constructed by comparing the identified amino acid sequences using MEGA7.0 and the neighbor-joining method (number of bootstrap replications were 1000). Protein domain architecture analysis was performed by a Conserved Domains Database search (https://www.ncbi.nlm.nih.gov/cdd).

## Genetic manipulations including knockout, complementation and overexpression

As mutations were identified in *VelB* (*MGG_01620*), genetic transformation was carried out to validate the roles of *VelB* in IPT resistance. Double-joint PCR was used to generate the knockout constructs of *M. oryzae* VelB [28]. Three knockout transformants for each gene were obtained, confirmed, and one transformant was randomly selected from each group for further analysis. To generate complemented transformants of the Δ*VelB* knockout transformant, a full *VelB* genomic region, including its upstream 2-kb region, was inserted into the plasmid pGTN for transformation. Overexpression transformants were obtained through the construct including the VelB-coding region and the 1.5 kb promoter region of histone 3 in the plasmid pTNHG. Genetic transformation was conducted by using PEG-mediated protoplast transformation. At the same time, other members of velvet family, i.e., *VeA* (*MGG_08556*), *VelC* (*MGG_14719*) and *VosA* (*MGG_00617*), as well as the VeA-interaction protein encoding gene *LaeA* (*MGG_01233*) were also knocked out and overexpressed to demonstrate the contribution of these genes in IPT resistance. *MoVelB* and *MoVeA* double knockout transformants, *MoVeA* and *MoLaeA* double knockout transformants and *MoVelB*, *MoVeA* and *MoLaeA* triple knockout transformants were constructed to analyze the relationship between MoVelB, MoVeA and MoLaeA.

## Stress measurement

To test sensitivity against different stresses, mycelial growth was assayed after incubation at 27˚C for 5 days on PDA plates and PDA amended with 0.7 M NaCl, 0.7 M KCl, 1.2 M sorbitol (SOR), 0.015% SDS (w/v), 1200 μg/mL Congo red (CR) or 1200 μg/mL calcofluor white (CFW), 6 mM $H_2O_2$.

## RNA preparation and qRT-PCR

Mycelia from the relevant strains were collected at indicated conditions and times, frozen rapidly in liquid nitrogen and stored at -80˚C until further use. Total RNA isolation was conducted by using TRIzol. cDNA was prepared using a HifairII 1st Strand cDNA Synthesis kit (YEASEN Biotech Co., Ltd) with oligo (dT). Reverse transcription quantitative PCR (RT-qPCR) was performed with ChamQTM SYBR qPCR Master Mix (Vazyme biotech co., Ltd) on a Bio-Rad CFX96 real-time PCR detection system. The comparative cycle threshold (CT) method was used for data analysis and relative fold difference was expressed as $2^{-\Delta\Delta CT}$ [29]. As an internal reference, primers for *MoActin* were used for each quantitative real-time PCR analysis. Primer sequences are shown in S2 Table.

## Yeast two-hybrid analysis

The full-length cDNA sequences of *VelB*, *VeA*, and *LaeA* were amplified to verify the potential interactions among *VelB*, *VeA*, and *LaeA* using Y2H assay. The cDNAs of *VelB* and *VeA* were respectively inserted into the *Eco*RI site of the pGBKT7 vector containing the GAL4 binding domain, and the *VeA* and *LaeA* cDNA was inserted into the *Eco*RI site of the pGADT7 vector containing the yeast GAL4 activation domain. The plasmid pairs of pGBKT7-VelB/pGADT7-VeA and pGBKT7-VelB/pGADT7-LaeA, pGBKT7-VeA/pGADT7-LaeA were co-transformed into the AH109 using the LiAc/Carry-DNA/PEG3350 transformation method [30]. The plasmid pairs of pGADT7/pGBKT7-53 and pGADT7/pGBKT7-Lam served as the positive and negative controls, respectively.

## Co-IP (Co-immunoprecipitation) assays

For in vivo Co-IP assays, constructs of targeted genes fused with different markers (GFP, Flag) were transformed in pairs into the wild-type H08-1a. The resulting transformants were confirmed by western blot analysis with monoclonal anti-Flag (66008, Proteintech, Hubei, China) and anti-GFP (66002, Proteintech, Hubei, China). The total proteins of strains containing the target protein alone or co-expressed proteins were extracted and further incubated with anti-Flag Affinity Gel (P2282, Beyotime, Shanghai, China) at 4°C overnight. Proteins eluted from the anti-Flag Affinity Gel were analyzed by western blot with the anti-GFP antibody.

## RNA sequencing

RNA sequencing was conducted on the Illumina HiSeq 4000 PE150 platform using 150 bp paired-end libraries with 500 bp inserts at Wuhan SeqHealth Technology Company. Transcriptome data quality was controlled using fastp (version 0.23.0) and over 35 million high-quality reads per sample were achieved. The RPKM (Reads per Kilobase per Million Reads) value used as a measure of gene expression, the gene was considered as a differentially expressed gene, when $\log_2$ (FoldChange ($\Delta VelB$\_RPKM/H08-1a\_RPKM)) $> 1$ or $< -1$ and p-value $< 0.05$ [31]. KEGG and GO analyses were performed using the DAVID Bioinformatics Resources online website (https://david.ncifcrf.gov/home.jsp) [32].

## Statistical analysis

Statistical differences were evaluated by one-way ANOVA with Duncan's Multiple Range tests in SPSS for Windows Version 19.0. Graphs were produced using GraphPad Prism 8. Statistical tests used for each reported experiment are detailed within figure legends.

## Supporting information

**S1 Table. Characteristics of *Magnaporthe oryzae* strains used in this study.**
(XLSX)

**S2 Table. Primers used in this study.**
(XLSX)

**S3 Table. Origin of mutants associated with *Magnaporthe oryzae*.**
(XLSX)

**S4 Table. Details of differentially expressed genes regulating secondary metabolism in $\Delta VelB$.**
(XLSX)

**S5 Table. Details of differentially expressed genes with monooxygenase activity in $\Delta VelB$.**
(XLSX)

**S1 Fig. Generation and identification of *MoVelB* knockout, complementation, overexpression transformants and determination of sensitivity to IPT.** (**A**) Gene replacement strategy of *MoVelB* and identification of $\Delta VelB$ transformants by PCR. (**B**) Sensitivity analysis of $\Delta VelB$ transformants to IPT. Data presented are the mean ± SD (n = 3). Bars followed by the same letter are not significantly different according to a LSD test at P = 0.01. (**C**) Gene complementation strategy of *MoVelB* and identification of *MoVelB* complementation transformants by RT-PCR. (**D**) Sensitivity analysis of *OEVelB* transformants to IPT. Data presented are the mean ± SD (n = 3). Bars followed by the same letter are not significantly different according to a LSD test at P = 0.01. (**E**) Detection of *MoVelB* expression in *OEVelB* transformants by RT-

qPCR. The *MoActin* gene was used as the internal reference for normalization. (**F**) Detection of variations of *MoVelB* in low resistant mutants.
(TIF)

**S2 Fig. Biological functions of velvet family gene *MoVelB*.** (**A**) Colony morphology of *MoVelB* knockout and complemented tansformants. (**B**) Melanin production in *MoVelB* knockout and complemented transformants. (**C**) Tolerance of *MoVelB* knockout and complemented transformants to different stresses. (**D**) Sensitivity of *MoVelB* knockout and complemented transformants to different fungicides. Strains were inoculated on different media with different stresses or fungicides at 27˚C for 5 days. IBP, FLU, IPR, TEB, AZO, BOS, and CAR indicate the fungicides iprobenfos, fludioxonil, iprodione, tebuconazole, azoxystrobin, boscalid, and carbendazim, respectively.
(TIF)

**S3 Fig. MoVeA regulated the sensitivity of $H_2O_2$, FLU and IPR.** (**A**) The H08-1a, $\Delta VeA$, $\Delta VosA$ and $\Delta VelC$ strains were incubated on PDA amended with different stress agents at 27˚C for 5 days and statistical analysis of the growth inhibition rate. (**B**) The H08-1a, $\Delta VeA$, $\Delta VosA$ and $\Delta VelC$ strains were incubated on PDA amended with different fungicides at 27˚C for 5 days and statistical analysis of the growth inhibition rate. Data presented are the mean ± SD (n = 3).
(TIF)

**S4 Fig. MoVelB-MoVeA-MoLaeA complex was involved in the regulation of sensitivity to IPT.** (**A**) The yeast two-hybrid (Y2H) assay revealed that MoVeA interacted directly with MoVelB. The plasmid pairs of pGADT7/pGBKT7-53 and pGADT7/pGBKT7-Lam served as the positive and negative controls, respectively. (**B**) Gene knockout strategy of *MoVeA* and identification of *MoVelB* and *MoVeA* double knockout transformants by PCR. (**C**) The yeast two-hybrid (Y2H) assay revealed that MoLaeA interacted directly with MoVeA, but not with MoVelB. (**D**) Gene knockout strategy of *MoLaeA* and identification of *MoVeA* and *MoLaeA* double knockout transformants, *MoVelB*, *MoVeA* and *MoLaeA* triple knockout transformants by PCR. (**E**) Sensitivity analysis of different velvet gene knockout transformants to IPT. Data presented are the mean ± SD (n = 3). Bars followed by the same letter are not significantly different according to a LSD test at P = 0.01.
(TIF)

**S5 Fig. VelB involved in IPT resistance through the regulation of secondary metabolism related genes.** (**A**) Expression levels of five secondary metabolism-related genes in $\Delta VelB$ and OE*VelB* strains by RT-qPCR. The *MoActin* gene was used as the internal reference for normalization. (**B**) GO analysis of DEGs in $\Delta VelB$ compared to H08-1a. (**C**) Expression heat map of differentially expressed monooxygenases encoding genes in $\Delta VelB$ compared to H08-1a.
(TIF)

## Author Contributions

**Conceptualization:** Fan-Zhu Meng, Liang-Fen Yin, Wei-Xiao Yin.

**Data curation:** Fan-Zhu Meng, Zuo-Qian Wang.

**Formal analysis:** Fan-Zhu Meng, Zuo-Qian Wang.

**Investigation:** Fan-Zhu Meng, Mei Luo, Wen-Kai Wei, Chao-Xi Luo.

**Methodology:** Fan-Zhu Meng, Mei Luo, Wen-Kai Wei, Chao-Xi Luo.

**Project administration:** Liang-Fen Yin, Wei-Xiao Yin.

**Supervision:** Wei-Xiao Yin.

**Validation:** Fan-Zhu Meng.

**Visualization:** Fan-Zhu Meng.

**Writing – original draft:** Fan-Zhu Meng.

**Writing – review & editing:** Guido Schnabel.

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
