## [Decision Letter · Decision Letter 0]

15 Mar 2023

Dear Dr. Luo,

Thank you very much for submitting your manuscript "The velvet family proteins mediate low resistance to isoprothiolane in Magnaporthe oryzae" for consideration at PLOS Pathogens. To start with please accept our apologies for the delay in processing this manuscript, which due to the difficulty in securing reviewers.

As with all papers reviewed by the journal, your manuscript was reviewed by members of the editorial board and by two independent reviewers. In light of the reviews (below this email), we would like to invite the resubmission of a significantly-revised version that takes into account the reviewers' comments. You should note the comments from reviewer 1 in particular. They make a number of useful suggestions in regards to providing evidence to support the conclusions drawn in this manuscript. These are suggestions and you can choose a different approach but the key here is to provide additional evidence in relation to the activity of the velvet complex in regulating IPT resistance.

We cannot make any decision about publication until we have seen the revised manuscript and your response to the reviewers' comments. Your revised manuscript is also likely to be sent to reviewers for further evaluation.

Sincerely,

Alex Andrianopoulos

Section Editor

PLOS Pathogens

Kasturi Haldar

Editor-in-Chief

PLOS Pathogens

orcid.org/0000-0001-5065-158X

Michael Malim

Editor-in-Chief

PLOS Pathogens

orcid.org/0000-0002-7699-2064

Reviewer's Responses to Questions

**Part I - Summary**

Reviewer #1: Focus of this manuscript entitled „The velvet family proteins mediate low resistance to isoprothiolane in Magnaporthe oryzae” where the authors performed an evolutionary adaptation experiment by exposing this fungus to antifungal drug IPT. They found three group of mutants with low and moderate group of resistance. MR-2 mutations were responsible for moderate level of resistance. However, for the low resistant mutants, interestingly they found a subunit of velvet complex MoVelB was mutated by an early stop codon. After finding this, they further studied other components of the velvet complex in Magnaporthe where they found all three components of the velvet complex, LaeA-VeA-VelB play important roles for IPT resistance. The authors conclude that the reason why the mutants of the velvet complex are resistant to IPT was down regulation of secondary metabolites or CYP450 genes. Main findings of this study are solid and sound. However, mechanism of resistance remains largely unknown, which is one of the flaws of this study.

Reviewer #2: In this manuscript, Meng et al., set out to determine the mechanisms of low-level resistance to the fungicide Isoprothiolane (IPT) in the rice blast fungus, Magnaporthe oryzae. The emergence of low-level resistance to IPT in blast populations is an ongoing problem in China, yet the mechanisms of low-level resistance are largely unknown, and therefore this study is novel and of significance. The manuscript is beautifully written, and the data seem solid, convincing, and mostly well-presented (however, some of the figures should be increased in size, as they were a little difficult to read in places). I noted only a few weaknesses in the manuscript. Firstly, precisely how the spontaneous IPT-resistant mutants were generated was not well described in the methods. As I understand it, hyphal plugs from the parental strain were inoculated on PDA followed by IPT-amended PDA. But it's not clear how many plugs were inoculated in total. Some clarification would be useful. Also, while I was comfortable with most of the conclusions in the paper, Line 249/250 states that "yeast two-hybrid assay showed that MoLaeA can directly interact with MoVeA, demonstrated that MoLae formed a heterotrimer complexes [sic]'. The yeast two-hybrid data do not, and could not, show heterotrimer formation, and the language should be changed. Ideally, these interactions would be confirmed by in vivo co-immunoprecipitation experiments, but I think this is beyond the scope of this manuscript.

**Part II – Major Issues: Key Experiments Required for Acceptance**

Reviewer #1: Regarding the resistance mechanism: Authors speculate that low level IPT mechanism is due to reduced secondary metabolite levels or CYP450 genes. In Figure 7D, expression of 12 genes were tested using qRT-PCR. The authors do not even name those genes what they are and why upregulation of these 5 genes make sense?

Have authors overexpress VelB, VeA and LaeA to see if their overexpression results in hypersensitivity to IPT. If their assumption “reduced secondary metabolite levels” causes low IPT resistance, then overexpression of these genes or at least two of them should cause an opposite phenotype to a deletion strain. The authors only overexpress MoVelB (Fig S1D), not VeA nor LaeA. Although they mentioned that the results are not significant, what I can see from FigS1D and its quantification that IPT resistance was reduced in MoVelB overexpression. Have authors quantify overexpression of MoVelB (gene expression) in comparison to WT H08-1a. Based on what expression level we call it overexpression? Is it 3 times more, 5 times? I am just wondering if similar effect can also be observed in VeA and LaeA overexpression. It would be interesting to see overexpression of these three mutants side by side with deletions. Furthermore, those genes which are drastically changed in Figure 7D should show opposite phenotype. Alternatively, the genes drastically change in opposite direction in velB knock out and overexpression strains should be the key genes of IPT resistance.

Line 244, “cytosolic protein LaeA……” how do the authors know that LaeA is a cytosolic protein? Do they have any evidence for this?

Figure 6E, only suggest that VeA and VelB form heterodimer via yeast two hybrid, which does not represent native forms of the protein interactions. Any CoIP would help to support their claims with these two proteins whether these interactions take place in vivo within the Magnaporthe.

Reviewer #2: n/a

**Part III – Minor Issues: Editorial and Data Presentation Modifications**

Reviewer #1: (No Response)

Reviewer #2: I found it difficult to read the labels on some of the figures due to their small size.

Line 254: I'm not sure about the use of 'deformation' to describe the disruption of a putative LaeA/VeA/VelB complex.

Fig.5 significance letter annotation are lowercase on G and E, and uppercase elsewhere.

PLOS authors have the option to publish the peer review history of their article (what does this mean?). If published, this will include your full peer review and any attached files.

Reviewer #1: No

Reviewer #2: No
---

## [Editor Report · Decision Letter 1]

24 May 2023

Dear Dr. Luo,

We are pleased to inform you that your manuscript 'The velvet family proteins mediate low resistance to isoprothiolane in Magnaporthe oryzae' has been provisionally accepted for publication in PLOS Pathogens.

Best regards,

Alex Andrianopoulos

Section Editor

PLOS Pathogens

Alex Andrianopoulos

Section Editor

PLOS Pathogens

Kasturi Haldar

Editor-in-Chief

PLOS Pathogens

orcid.org/0000-0001-5065-158X

Michael Malim

Editor-in-Chief

PLOS Pathogens

orcid.org/0000-0002-7699-2064
---

## [Editor Report · Acceptance letter]

2 Jun 2023

Dear Dr. Luo,

We are delighted to inform you that your manuscript, "The velvet family proteins mediate low resistance to isoprothiolane in Magnaporthe oryzae," has been formally accepted for publication in PLOS Pathogens.

Best regards,

Kasturi Haldar

Editor-in-Chief

PLOS Pathogens

orcid.org/0000-0001-5065-158X

Michael Malim

Editor-in-Chief

PLOS Pathogens

orcid.org/0000-0002-7699-2064